# Lymphatic filariasis in 2016 in American Samoa: Identifying clustering and hotspots using non-spatial and three spatial analytical methods

Kinley Wangdi[1]*, Meru Sheel[2], Saipale Fuimaono[3], Patricia M. Graves[4], Colleen L. Lau[1,5]

1 Department of Global Health, Research School of Population Health, College of Health and Medicine, Australian National University, Acton, Canberra, Australia, 2 National Centre for Epidemiology and Population Health, Research School of Population Health, College of Health and Medicine, Australian National University, Acton, Canberra, Australia, 3 American Samoa Department of Health, Pago Pago, American Samoa, 4 College of Public Health, Medical and Veterinary Sciences and Australian Institute of Tropical Health and Medicine, James Cook University, Cairns, Australia, 5 School of Public Health, Faculty of Medicine, The University of Queensland, Herston, Australia

* kinley.wangdi@anu.edu.au

**Data Availability Statement:** All relevant data are within the paper. We are unable to provide individual-level antigen and antibody prevalence

## Abstract

### Background

American Samoa completed seven rounds of mass drug administration from 2000–2006 as part of the Global Programme to Eliminate Lymphatic Filariasis (LF). However, resurgence was confirmed in 2016 through WHO-recommended school-based transmission assessment survey and a community-based survey. This paper uses data from the 2016 community survey to compare different spatial and non-spatial methods to characterise clustering and hotspots of LF.

### Method

Non-spatial clustering of infection markers (antigen [Ag], microfilaraemia [Mf], and antibodies (Ab [Wb123, Bm14, Bm33]) was assessed using intra-cluster correlation coefficients (ICC) at household and village levels. Spatial dependence, clustering and hotspots were examined using semivariograms, Kulldorf's scan statistic and Getis-Ord Gi* statistics based on locations of surveyed households.

### Results

The survey included 2671 persons (750 households, 730 unique locations in 30 villages). ICCs were higher at household (0.20–0.69) than village levels (0.10–0.30) for all infection markers. Semivariograms identified significant spatial dependency for all markers (range 207–562 metres). Using Kulldorff's scan statistic, significant spatial clustering was observed in two previously known locations of ongoing transmission: for all markers in Fagali'i and all Abs in Vaitogi. Getis-Ord Gi* statistic identified hotspots of all markers in Fagali'i, Vaitogi,

data and demographic data because of the potential for breaching participant confidentiality. The communities in American Samoa are very small, and individual-level data such as age, sex, and village of residence could potentially be used to identify specific persons. For researchers who meet the criteria for access to confidential data, the data are available on request from the Human Ethics Officer at the Australian National University Human Research Ethics Committee, email: human. ethics.officer@anu.edu.au. Protocol number 2016/ 482.

**Funding:** This work received financial support from the Coalition for Operational Research on Neglected Tropical Diseases (COR-NTD), which is funded at The Task Force for Global Health primarily by the Bill & Melinda Gates Foundation (Grant number OPP1053230), the United Kingdom Department for International Development, and the United States Agency for International Development through its Neglected Tropical Diseases Program. M.S. was funded by a Westpac Research Fellowship. C.L.L. was funded by Australian National Health and Medical Research Council Fellowships (APP1109035 and APP1158469). The funders had no role in study design, data collection and analysis, decision to publish, or preparation of the manuscript. This work was supported in whole or in part, by the Bill & Melinda Gates Foundation [Grant Number OPP1053230]. Under the grant conditions of the Foundation, a Creative Commons Attribution 4.0 Generic License has already been assigned to the Author Accepted Manuscript version that might arise from this submission.

**Competing interests:** The authors have declared that no competing interests exist.

and Pago Pago-Anua areas. A hotspot of Ag and Wb123 Ab was identified around the villages of Nua-Seetaga-Asili. Bm14 and Bm33 Ab hotspots were seen in Maleimi and Vaitogi-Ili'ili-Tafuna.

## Conclusion

Our study demonstrated the utility of different non-spatial and spatial methods for investigating clustering and hotspots, the benefits of using multiple infection markers, and the value of triangulating results between methods.

## Author summary

Lymphatic filariasis is a parasitic infection caused by thread-like filarial nematodes and transmitted by mosquitoes. Lymphatic filariasis was endemic in American Samoa and seven rounds of mass drug administration were distributed between 2000 and 2006. Routine blood surveys in 2011 and 2015 did not identify any evidence of ongoing transmission. However, research studies conducted at around the same time showed evidence of residual hotspots and ongoing transmission, which was confirmed by both school-based and community-based surveys in 2016. This study analysed data from the 2016 community survey to identify clusters and hotspots using both non-spatial and spatial analytical methods. The findings confirmed previously known locations of ongoing lymphatic filariasis transmission in American Samoa and identified other potential hotspots that warrant further investigation. We demonstrated the utility of different non-spatial and spatial methods for investigating clustering and hotspots, and different information provided by each method. Noting the added value of these methods, they could potentially be considered as additional tools for improving lymphatic filariasis surveillance and optimising operational activities for elimination programmes, particularly for identifying areas of ongoing transmission or resurgence that may not be identified through current surveillance strategies.

## Introduction

Lymphatic filariasis (LF) is a neglected tropical disease, with an estimated 51.4 million people infected in tropical and subtropical areas globally in 2018 [1]. LF is caused by three species of thread-like filarial nematodes, *Wuchereria bancrofti*, *Brugia malayi* and *B. timori* [2]. Infection occurs when filarial parasites are transmitted to humans by mosquitoes including *Aedes*, *Anopheles*, *Culex* and *Mansonia* species [3]. Infection may cause damage to the lymphatic system and result in chronic disability, including lymphoedema, elephantiasis and scrotal hydrocoele [4]. In endemic countries, LF has major social and economic impacts with an estimated annual cost of US$1 billion [5].

The World Health Organization (WHO) Global Programme to Eliminate Lymphatic Filariasis (GPELF) aims to eliminate LF as a public health problem using a two-pronged approach: (i) to interrupt transmission of LF by conducting mass drug administration (MDA) with antihelminthic medications annually to entire communities in endemic regions, and (ii) morbidity management and disability prevention for people with chronic complications [6,7]. Interventions conducted through GPELF are estimated to have prevented or treated more than 97 million cases and averted more than US$100 billion in economic losses over the lifetime of those affected [8,9].

In 1999, the Pacific Programme to Eliminate LF (PacELF) was formed to manage LF elimination in the 16 endemic Pacific Island Countries and Territories (PICTs) in the South Pacific region, including American Samoa [10]. In this region, *W. bancrofti* is transmitted by many vector genera including *Aedes*, *Anopheles* and *Culex*. Amongst the PICTs, Cook Islands, Niue, the Marshall Islands, Palau, Tonga, Vanuatu, Wallis and Futuna, and Kiribati have successfully achieved targets and received validation by WHO as having eliminated LF as a public health problem [10]. In American Samoa, LF is caused by *W. bancrofti* and the main vector is *Aedes polynesiensis* (day-biting). Other local vectors include *Ae. samoanus* (night-biting), *Ae. tutuilae* (night-biting), and *Ae. upolensis* (day-biting) [11,12].

American Samoa has made efforts to eliminate LF through two MDA programs. Firstly, in 1963 and 1965 [13] with repeated doses of diethylcarbamazine (DEC), and secondly as part of PacELF [14], seven MDA rounds were distributed between 2000 and 2006 using single annual doses of DEC plus albendazole [14]. Transmission assessment surveys (TAS) in 6–7 year old children passed the recommended threshold of antigen (Ag) prevalence (upper 95% confidence interval [CI] of <1%) set by WHO for areas with *W. bancrofti* and *Aedes* vectors [15] in 2011–2012 (TAS-1) [16] and 2015 (TAS-2) [17]. Despite these successes, operational research studies conducted outside of programmatic activities detected residual hotspots and ongoing transmission in American Samoa in 2010, 2014 and 2016 [18,19]. In the context of these studies, the term 'hotspot' was used to refer to localised areas where Ag prevalence was significantly >1%, and higher compared to the rest of the study area, and the term "resurgence" was used to indicate significant increase in infection prevalence to levels above target thresholds. In the 'TAS Strengthening Survey' conducted in 2016, where a community-based cluster survey was undertaken in parallel with TAS-3 conducted in all elementary schools, both surveys confirmed the resurgence of LF. The study demonstrated that the community-based survey of older age groups (≥8 years) was more sensitive than TAS of 6–7 year-old children for identifying signals of ongoing transmission, including hotspots identified in 2010 and 2014 studies: Fagali'i village in the far north-west of Tutuila island and a group of three villages (Ili'Ili, Vaitogi and Futiga) on the south coast [20,21].

Spatial analytical methods and geographic information systems (GIS) have increasingly been used in public health [22–26]. Hotspot and cluster analyses are examples of spatial statistical methods that can be used to assess geographic variation in disease risk and/or occurrence of a disease in excess of what is expected within a geographic location. As countries near LF elimination targets, identifying the most practical and robust tools for LF surveillance will aid in finding the last reservoirs of infection. Spatial stratification of infection risk and reliable identification of hotspots could potentially be used to strengthen surveillance, inform more precise targeting of interventions, and maximise the chances of achieving elimination. In American Samoa, our previous work using spatial analyses identified clustering of Ag-positive adults in 2010 [19]. In this paper, we use the results of the 2016 community-based survey in American Samoa to investigate the spatial epidemiology of LF when there was strong evidence of resurgence (after adjusting for survey design, age and sex, the estimated Ag prevalence in 2016 was 6.2% (95% CI 4.5–8.6%) in residents aged >8 years [18,19]). This study aimed to identify clustering and hotspots of LF Ag, microfilariae (Mf), and antibodies (Ab) using both non-spatial and spatial analytical methods, and compare the results between different methods.

## Materials and methods

### Ethics statement

This study was approved by the American Samoa Institutional Review Board and the Human Research Ethics Committee at the Australian National University (protocol number 2016/

482). The American Samoa Department of Health and the American Samoa Community College were local collaborators and provided local guidance and logistical support. The permission to visit villages was granted by the Department of Samoan Affairs. All field activities were carried out in a culturally appropriate and sensitive manner with bilingual local field teams, and with verbal approval sought from village chiefs/ mayors prior to conducting the community surveys. A signed informed consent to collect demographic data and blood samples was obtained from adult participants or from parents/guardians of the participants <18 years, along with verbal assent from minors [21]. Surveys were conducted in English or Samoan depending on the participants' preference. For the 2016 field study, the Institutional Review Board of the U.S. Centers for Disease Control and Prevention (CDC) determined the CDC to be a non-engaged research partner.

## Study site

The study was carried out in American Samoa, a United States territory located in the South Pacific ($14.2710^0$ South, $170.1322^0$ West), with a total area of 205.8 $km^2$ made up of five main islands and two coral atolls. The ~70 villages range in geographic size from 0.16 $km^2$ (Atu'u) to 14.9 $km^2$ (Si'ufaga). Ninety percent of the population lives on Tutuila, the largest island. In 2017, the total estimated population of American Samoa was 60,300 [27].

## Data sources

Data for this study were obtained from the community-based survey carried out in 2016 [21]. Briefly, the study used a two stage community-based probability cluster survey, where clusters (primary sampling units [PSUs]) were selected in stage one and households in stage two. PSUs with <2000 residents were created; this required bigger villages to be divided into segments and very small adjacent villages were grouped. Thirty PSUs (from 28 villages) were randomly selected from a total of 70 villages; results of LF seroprevalence from these PSUs have previously been described by Sheel *et al* [21] and Lau *et al* [19]. In addition, two villages (Fagali'i and Futiga) that were previously identified as hotspots in 2010, and confirmed in 2014, were purposively sampled as additional PSUs and results reported by Lau *et al* [19]. This study includes results from 750 households in all 32 PSUs (across 30 villages). For each PSU, a population proportionate number of households were randomly selected from the geo-referenced lists of houses and buildings provided by the American Samoa Department of Commerce [28]. In Fagali'i, volunteers were included from non-randomly selected households; the demographics and seroprevalence were similar in the randomly selected and volunteer participants, so results were combined for analyses. All household members aged ≥8 years were invited to participate.

Household GPS coordinates were recorded using the LINKS electronic database system developed by the Task Force for Global Health [29]. Of 2671 participants included in the analyses, accurate GPS locations (to within 25m) were available for 2630 (98.5%) persons in 750 households, either from GPS coordinates recorded using the LINKS system or from hard copies of fieldwork maps. For 41 (1.5%) participants, exact GPS locations were not available, and the coordinates of the village centroid were used. Village shapefiles were downloaded from DIVA-GIS website (https://www.diva-gis.org/Data).

## Infection markers

During the 2016 household survey, 200μL of heparinized finger prick blood samples were collected from each participant and tested for circulating filarial Ag using the Alere Filariasis Test Strip (FTS) [21]. Dried blood spots (DBS) were prepared by spotting 60uL of blood (10 μL per

extension x 6 extensions) onto filter papers (Cellabs, Sydney, Australia), dried and stored at -20˚C, and shipped to the US Centers for Disease Control and Prevention for anti-filarial Ab (Bm14, Bm33, Wb123) testing using multiplex bead assays [17,30]. For all Ag-positive individuals, additional heparinised venous blood samples were collected for microscopic examination for Mf where possible. Mf slides were prepared using 60uL of blood and stained with Giemsa [9]. Those who were Ag-negative were deemed as Mf-negative.

Mf positivity represents active infection and infectiousness. Ag indicates the presence of live or dead adult filarial worms in the lymphatic system, and may persist for months or years after treatment [31,32]. The Wb123 Ab was identified by a library generated from *W. bancrofti* L3 larval stages [33]; it appears in the early stages of infection and persists for long periods after infection; antibody dynamics post-infection and post-treatment are not well understood, and may differ between adults and children [34]. The Bm14 Ab was identified from a cDNA library screened using sera from microfilariaemic people [35,36]. Bm33 Ab was also identified in a *B. malayi* cDNA library as a major cross-reacting immunogen in *W. bancrofti* [37]. Bm14 and Bm33 Abs may persist long (many years) after infection, but exact duration is not well known.

## Non-spatial analysis

Intra-cluster correlation coefficient (ICC or rho [$\rho$]) was used to provide a measure of the degree of clustering for Ag, Mf and each Ab at the household and village levels. ICC within households was estimated using mixed effects logistic regression, with age and sex included as fixed effects. The ICC provides a measure of how strongly similar observations are clustered across the households and villages. As the variable of interest (e.g. Ag-positive) becomes more homogeneous at the village or household levels (i.e. stronger clustering), the ICC tends to approach one. Conversely, as variables become more heterogeneous, the ICC approaches zero. ICC does not take into account the spatial distribution or geographic locations of households or villages.

## Spatial analyses

i) Spatial dependence

Spatial dependence (or spatial autocorrelation) is the concept that observations closer together in space have a tendency to be more similar than those that are further apart. A semivariogram is a graphical representation of semivariance on the y-axis as a function of the distance between pairs of observations (x-axis). A semivariogram is defined by three parameters: the sill- the semivariance at which the variogram plateaus (indicative of statistical significance vs no plateauing if not significant); the nugget- the value at which semivariance intercepts the y-axis (represents spatial variability or measurement error); and the range- the distance between the y-axis and point at which the sill first flattens out (represents the size of geographical clusters). Partial sill is the sill minus the nugget. The proportion of the variation attributed to spatial structure (geographical proximity) can be calculated by dividing the partial sill by the sum of partial sill and nugget (equal to dividing nugget by sill) [38,39]. The ratio of nugget to sill provides a measure of the strength of spatial dependence, where ratios of <25%, 25–75%, and >75% indicate strong, moderate, and weak dependence, respectively.

For each infection marker, spatial dependence of positive results at the household level was investigated using semivariograms in the statistical software R, using the *variog* function in geoR package version 2.14.1 (The R Foundation for Statistical Computing). The method assigns a series of intervals ("lags") within the sampled area up to the maximum range, and

calculates semivariance (as a measure of relatedness between observations) of the outcome of interest for all pairs of observations within each lag. Twenty lags were used in this study. The R package "*plot.geodata*" was used to determine the maximum distance between pairs of survey points for x and y coordinates; half of the shorter of these two distances was used as the maximum range. The maximum distance between survey points were 0.06 and 0.1 decimal degrees for x and y coordinates, respectively. Therefore, 0.03 decimal degrees (~3.3km) was used as the maximum range for semivariograms. Outputs in decimal degrees were converted to metres based on the assumption of one degree at the equator being equivalent to 111km; reported cluster sizes in metres are thus slight overestimates because American Samoa is located approximately 1500km south of the equator.

Outputs from semivariograms are *global* (rather than local) measures of the degree of clustering (i.e. overall measures of spatial structure in the data) and do not provide information on the geographic locations of clusters. The global significance of clusters of infection markers was also assessed using Moran's I Statistics.

ii) Spatial clustering and hotspots

Two methods of *local* spatial analysis, the Kulldorff's scan statistic and the Getis-Ord Gi* statistic, were used to identify locations of significant clusters and hotspots of Ag, Mf, and each Ab. The Kulldorff's scan statistic was used to determine the geographic distribution of clusters of high prevalence of infection markers, while Getis-Ord Gi* statistic was used to test the statistical significance of hotspots and to determine the spatial dependence between observations (see below).

In this study, significant SaTScan results are referred to as clusters. Kulldorff's scan statistic was determined using SaTScan software [40] which uses moving scanning windows (circular or elliptical) of varying sizes to estimate the probability that the frequency of positive individuals within a window is in excess of what is expected by chance. SaTScan takes into account the observed number of positive and negative individuals inside and outside the windows, calculates the relative risk (RR) of positive cases within each window, and reports locations of windows (clusters) where there is a statistically significantly higher proportion of positive cases within the window [41,42]. SaTScan was set to include a maximum of 25% of the observations within circular windows, and a Bernoulli model was used because the outcome variables of interest were binomial (positive and negative results). For each location, the window size with the highest log likelihood ratio (LLR) was considered the most probable cluster, i.e. the cluster that is least likely to have occurred randomly. The SaTScan output for statistically significant clusters includes the location of the centre of the scanning window, the radius of the scanning window, the number of observed and expected positives within the circle, relative risk, LLR, and *p* value. The statistically significant clusters were explored at Monte Carlo replications of 999 to ensure adequate power for defining clusters, and were considered significant at *p* <0.05. The clusters were ranked based on the LLR, and those with higher LLR were associated with higher relative risk. The locations of surveyed households and significant clusters identified by SaTScan were mapped using ArcMap 10.5.1 (ESRI, Redlands, CA).

The significant results from Getis-Ord Gi* analyses are referred to as hotspots. Hotspot analysis was conducted using the Getis-Ord Gi* statistic in ArcMap 10.5.1 (ESRI, Redlands, CA). The Gi* statistic is a z-score that identifies areas of higher or lower values by comparing them to a normal probability distribution, and provides a measure of the local concentration of positive individuals. In this study, each location with a positive test result was assigned a value "1" and a negative result a "0". We used the Fixed Distance Band for the conceptualization of spatial relationships; this statistic compares spatial dependency of positive individuals between locations to identify hotspots and coldspots. A statistically significantly large positive

z-score signifies a large number of positives in a local area (hotspot), while a large negative z-score signifies a low number of positives (cold spot) [43]. The Getis-Ord Gi* statistic was used to classify surveyed locations into hotspots and coldspots with 90%, 95%, and 99% confidence.

## Results

### Demographics of participants

The 2016 household survey included 2710 participants from 32 PSUs in 30 villages [20,21]. Of these, 12 (0.4%) participants who were aged <8 years, 22 (0.8%) who had missing or invalid laboratory test results (13 for Ag and nine for Abs), and five (0.2%) who had missing household geocoordinates were excluded from this study. The final dataset used for analyses included 2671 participants with a mean age of 33.5 years (range 8–93), and 54.7% (n = 1462) were female. The study included 750 households at 730 unique survey locations; some households shared the same structure, e.g. apartment buildings. The median number of participants per household was three (range 1–20), and the median number of residents per household was five (range 1–25). Children surveyed in TAS-3 (conducted in parallel with the household survey) were not included in this study because data on their household locations were not collected (except for the nine Ag-positive children).

### Prevalence of Ag, Mf, and antibodies

Results of population estimates of the prevalence of infection markers (Ag, Mf, and Abs), adjusted for age, sex, and survey design, have previously been reported [20,21]. Of the 2671 individuals included in this study, 5.1% (n = 135) were positive for Ag, 13.1% (n = 350) for Bm14 Ab, 25.6% (n = 684) for Wb123 Ab, and 45.9% (n = 1219) for Bm33 Ab. Results for Mf were unavailable for 21/135 (15.6%) Ag-positive participants either due to not participating in follow-up testing or due to insufficient blood available for Mf slides. Results for Mf slides were available for 114 (84.4%) of the Ag-positive participants, of which 34 were Mf-positive. For analyses, all Ag-negative persons (n = 2536) were assumed to be Mf-negative. Using a denominator of 2650 (2671 minus 21 Ag-positive participants for whom slide results were not available), Mf prevalence was 1.3% (no. of Mf positives = 34). Crude Mf, Ag, and Ab (Bm14, Wb123, and Bm33) prevalence by age and gender are presented in Fig 1.

### Number of household locations with Ag, Mf, and Ab positive persons

Of the 750 households, at least one positive person was identified in 92 (12.3%) households for Ag, 25 (3.3%) for Mf, 407 (54.3%) for Wb123 Ab, 244 (32.5%) for Bm14 Ab, and 581 (77.5%) for Bm33 Ab. S1 Table provides further details on the number of households with one, two, and more than two positive persons for each of the infection markers.

Of the 730 unique survey locations, at least one positive person was identified in 91 (12.5%) locations for Ag (Fig 2A), 25 (3.4%) for Mf (Fig 3A), 402 (55.1%) for Wb123 Ab (Fig 4A), 243 (33.3%) for Bm14 Ab (Fig 5B), and 573 (78.5%) for Bm33 Ab (Fig 6A). S2 Table provides details on the number of unique survey locations with one, two, and more than two positive persons for each of the infection markers.

### Non-spatial analysis

The ICC was lower at the village level compared to the household level for all infection markers (Table 1). At the household level, the highest ICC (strongest clustering) was observed for Mf (0.69) followed by Ag (0.59), Bm14 Ab (0.33), Wb123 Ab (0.27), and Bm33 Ab (0.20). At the

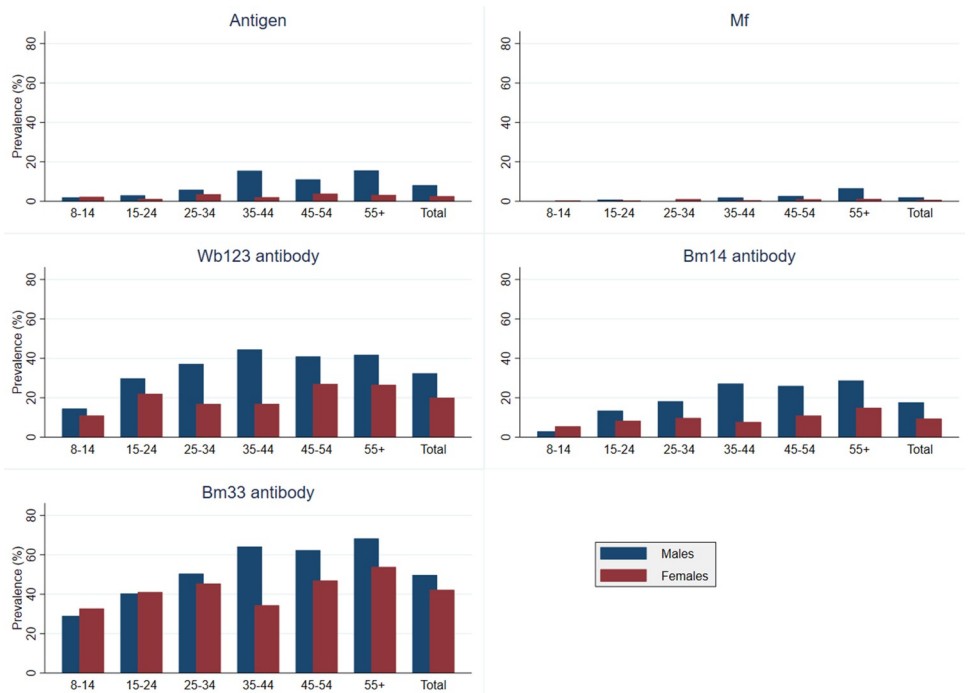

**Fig 1. Seroprevalence of lymphatic filariasis infection markers stratified by age and gender, American Samoa 2016.**

village level, similar patterns were seen, with the highest ICC for Mf (0.30) followed by Ag (0.17) and Abs (0.10–0.17) (Table 1).

## Spatial analysis

Semivariograms showed statistically significant spatial dependency (plateauing of semivariogram) for all infection markers (Table 2). The average size of a cluster for Ag was 562 metres and the proportion of the variation explained by geographical proximity was 14%. The cluster size for Mf was 207 metres, with 26% of variance explained by spatial dependency. Clusters sizes for Wb123, Bm14 and Bm33 Abs were 397, 548 and 220 metres, and 37%, 21% and 40% of variance were explained by geographical proximity, respectively (Fig 7 and Table 2). The nugget to sill ratio indicated moderate spatial dependence for Mf, Wb123 Ab and Bm33 Ab, and weak dependence for Ag and Bm14 (Table 2).

Moran's I Statistics showed there was global clustering (S3 Table). Using Kuldorff's scan statistic (SaTScan), significant clustering of all infection markers was identified around Fagali'i village (cluster 1) in the north west, an area of high Ag prevalence identified by our previous studies from 2010 and 2014 (Figs 2B, 3B, 4B, 5B and 6B). RR was highest for Mf (41.46, p<0.0001), followed by Ag (16.10, p<0.0001), Bm14 Ab (6.13, p<0.0001), Wb123 Ab (3.43, p<0.0001), and Bm33 Ab (2.13, p<0.0001) (Table 3). The Ag cluster had the smallest radius (0.44 km) while clusters were larger for Mf (1.85 km) and all three Abs (2.31 km) (Table 3). Significant clustering of Wb123, Bm14 and Bm33 Abs (RR = 2.68, p<0.001; 3.15, p = 0.0025 and 1.41, p<0.001) were identified in Vaitogi (cluster 2), a previously known area of high Ag prevalence (Figs 4B, 5B and 6B). Clustering of Bm33 Ab was also identified in Ili'ili-Tafuna (cluster 3) adjacent to Vaitogi (RR = 1.64, p = 0.015) and in the Pago Pago-Anua area (cluster 4, RR = 1.69, p = 0.044, Fig 6B).

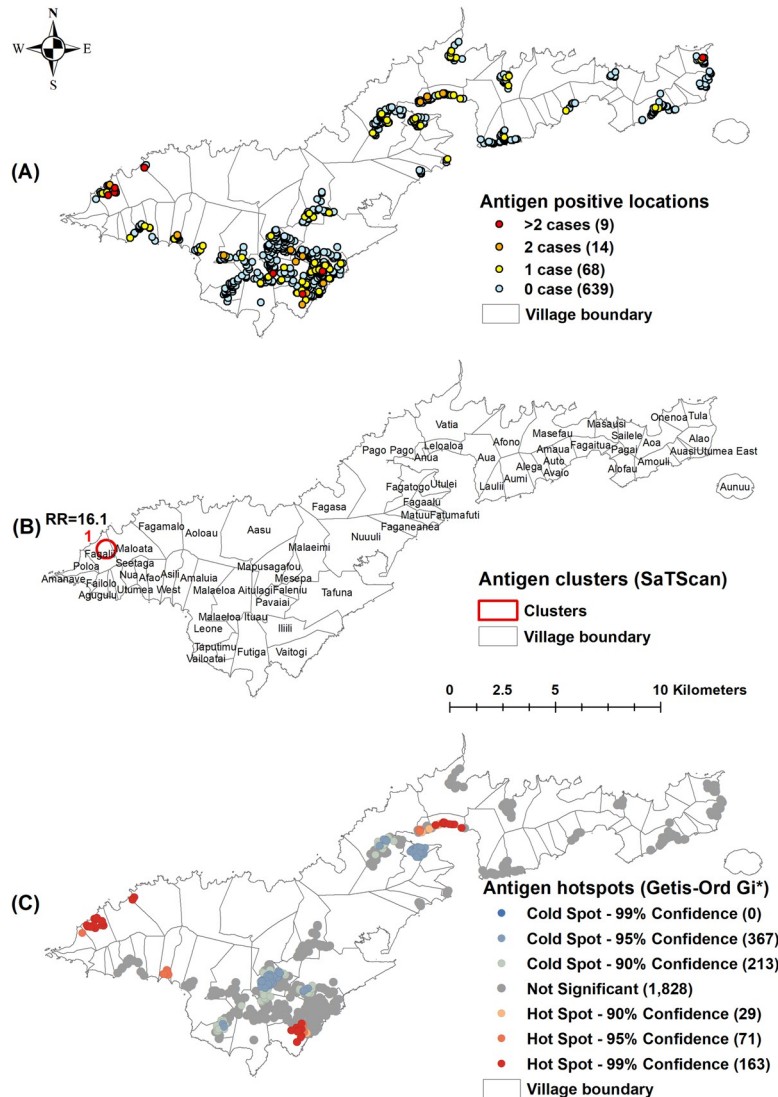

**Fig 2.** A) Spatial distribution of lymphatic filariasis antigen in survey locations, B) Spatial clusters of antigen positive survey locations (SaTScan) with relative risk (RR), C) Hotspots of antigen positive survey locations (Getis-Ord Gi*), American Samoa 2016. Base layers from (https://www.diva-gis.org/Data).

The Getis-Ord Gi* statistic identified hotspots of all infection markers at both Fagali'i and Vaitogi, in the vicinity of clusters 1 and 2 identified by SaTScan (Figs 2C, 3C, 4C, 5C and 6C). Hotspots of Wb123 and Bm33 Ab were identified in Ili'ili-Tafuna, around the Bm33 Ab cluster 3 identified by SaTScan. Hotspots of Ag, Mf, Wb123 and Bm33 Abs (but not Bm14 Ab) were identified in the Pago Pago-Anua-Leloaloa area, near where SaTScan identified Bm33 Ab cluster 4. Getis-Ord Gi* analyses also highlighted a large hotspot of Bm33 Ab in the Ili'ili-Vaitogi-Tafuna area (Fig 6C), overlapping clusters 2 and 3 identified by SaTScan (Fig 6B). In addition, Getis-Ord Gi* highlighted hotspots that had not been identified by our previous studies or by SaTScan in this study, the two most notable being a hotspot of Ag and Wb123 Ab around the villages of Nua-Seetaga-Asili in the south west (Figs 2C and 4C), and a hotspot of Wb123, Bm14, and Bm33 Abs at the inland village of Maleimi (Figs 4C, 5C and 6C).

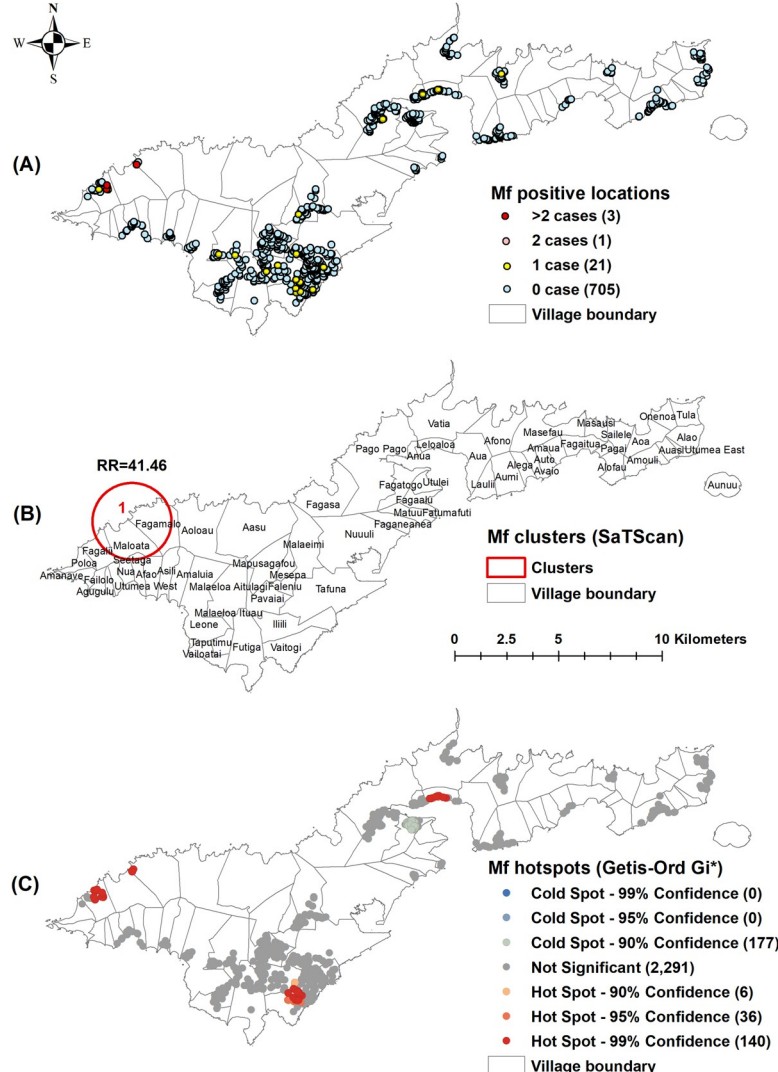

**Fig 3.** A) Spatial distribution of Mf in survey locations, B) Spatial clusters of antibody positive survey locations (SaTScan) with relative risk (RR), C) Hotspots of antibody positive survey locations (Getis-Ord Gi*), American Samoa 2016. Base layers from (https://www.diva-gis.org/Data).

## Prevalence of Ag, Mf, and antibodies within SaTScan clusters

The prevalence of Ag, Mf, and Abs within each of the clusters identified by SaTScan are summarised in Fig 8, and the overall prevalence in the study population is included for comparison (Fig 8A). Fig 8 shows that there was a generally higher prevalence of Ag, Mf, and Ab positives within all clusters compared to the overall prevalence in the study population. SaTScan identified significant clustering of all infection markers in the Fagali'i area (cluster 1); although the exact location and size of the cluster varied between the infection markers (Figs 2B, 3B, 4B, 5B and 6B). Prevalence of all infection markers was strikingly high in cluster 1, with >50% Ag-positive and >30% Mf-positive in the Ag and Mf clusters (Fig 8B and 8C), and >60% positive for all antibodies in the clusters identified using any infection marker (Fig 8D, 8E, and 8F).

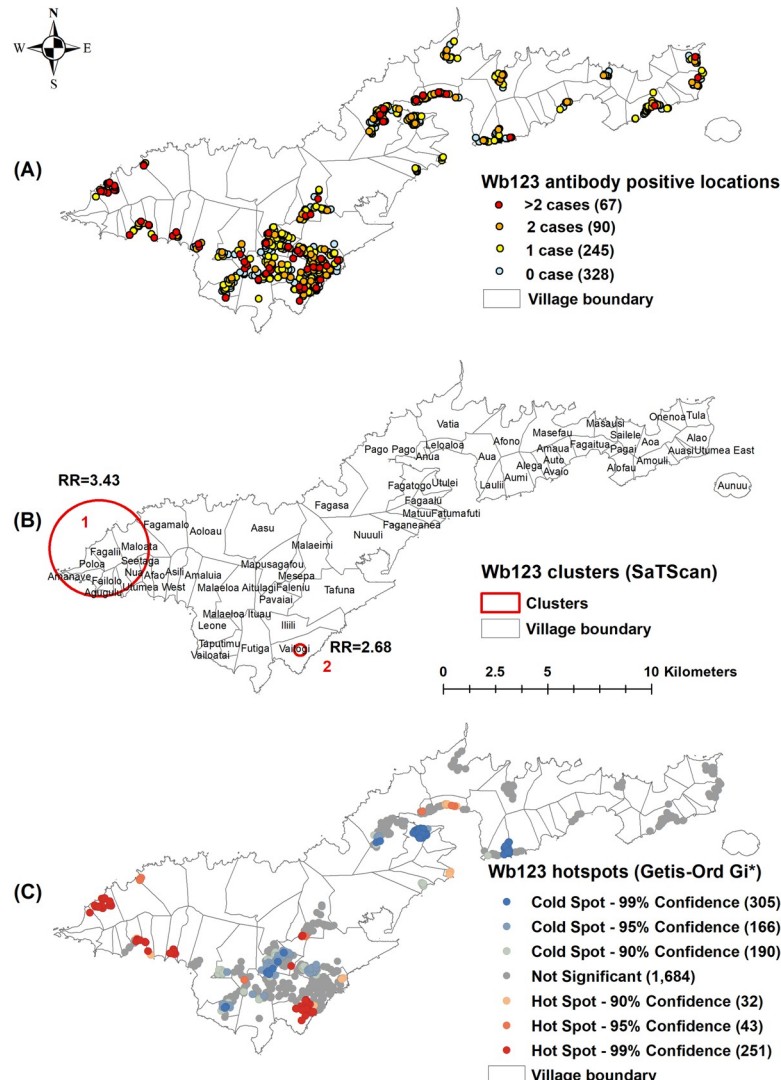

**Fig 4.** A) Spatial distribution of Wb123 antibody in survey locations, B) Spatial clusters of antibody positive survey locations (SaTScan), C) Hotspots of antibody positive survey locations (Getis-Ord Gi*), American Samoa 2016. Base layers from (https://www.diva-gis.org/Data).

SaTScan identified clusters of Wb123, Bm14 and Bm33 Abs in approximately the same area in Vaitogi (cluster 2); although SaTScan did not identify any Ag or Mf clusters in this area, the Ag prevalence was 16.7% in the Wb123 Ab cluster (Fig 8D), 15.1% in the Bm14 Ab cluster (Fig 8E), and 7.2% in the Bm33 Ab cluster (Fig 8F), compared to overall Ag prevalence of 5.1% in the study population (Fig 8A). Similarly, Mf prevalence in Vaitogi was 5.7% in the Bm14 Ab cluster (Fig 8E) and 6.3% in the Wb123 Ab cluster (Fig 8E), more than four times higher than the Mf prevalence of 1.3% in the study population (Fig 8A).

SaTScan also identified two additional clusters of Bm33 Ab which were not apparent with the other infection markers. In the Bm33 Ab cluster in the Ili'ili-Tafuna area (cluster 3) (Fig 8F), Ag and Mf prevalence were 8.7% and 2.2%, respectively, and in the Bm33 Ab cluster in the Pago Pago-Anua (cluster 4), Ag and Mf prevalence were 1.6% and 0%, respectively (lower than the overall prevalence in the population).

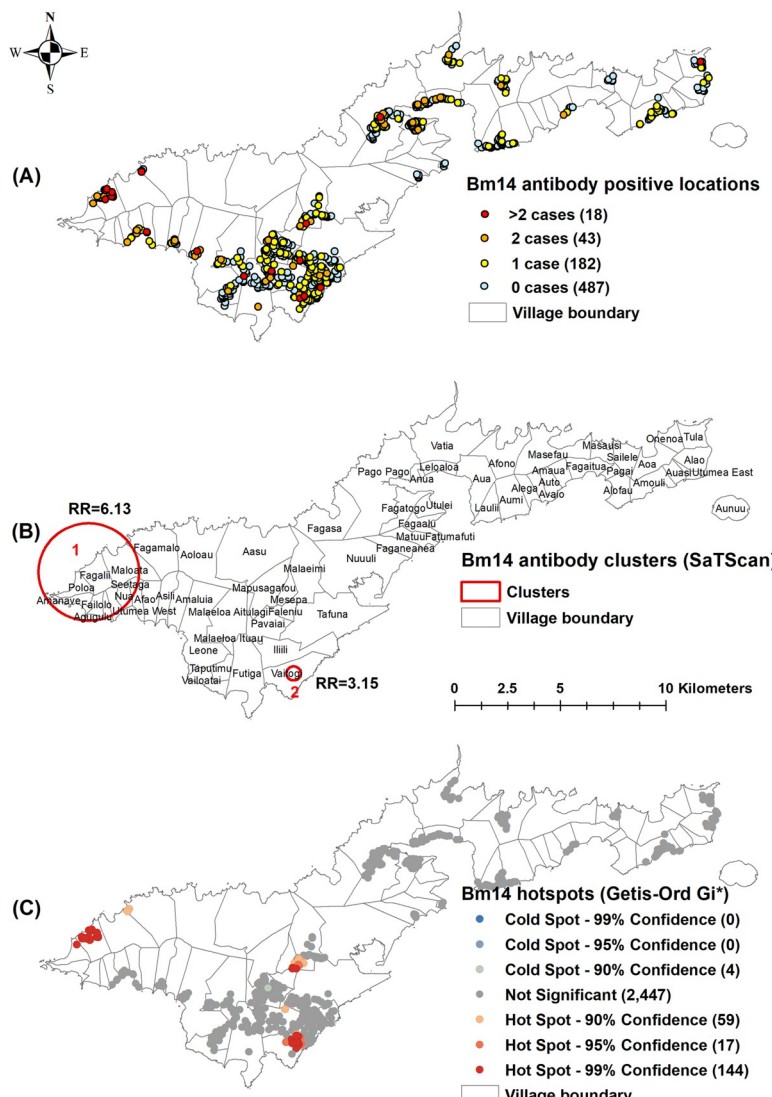

**Fig 5.** A) Spatial distribution of Bm14 antibody in survey locations, B) Spatial clusters of antibody positive survey locations (SaTScan), C) Hotspots of antibody positive survey locations (Getis-Ord Gi*), American Samoa 2016. Base layers from (https://www.diva-gis.org/Data).

## Discussion

Our study applied non-spatial and three different spatial analytical methods to 2016 community survey data from American Samoa [20,21], and identified clustering and hotspots of five LF infection markers (Ag, Mf, and Bm14, Wb123, and Bm33 Abs). Results varied between methods and infection markers, with each method providing different but complementary information about clustering and hotspots. ICC (a non-spatial measure) showed more intense clustering at the household level compared to the village level. Semivariograms (a global measure of spatial dependency) identified significant spatial dependency for all infection markers, with different cluster size for each marker. SaTScan (a local spatial statistic) identified a small number of clusters, including in locations of the two hotspots previously identified in 2010 and 2014 [18,19]. Getis Ord Gi* (another local spatial statistic) identified hotspots in the areas around the previously known hotspots, and appeared the most sensitive of the methods

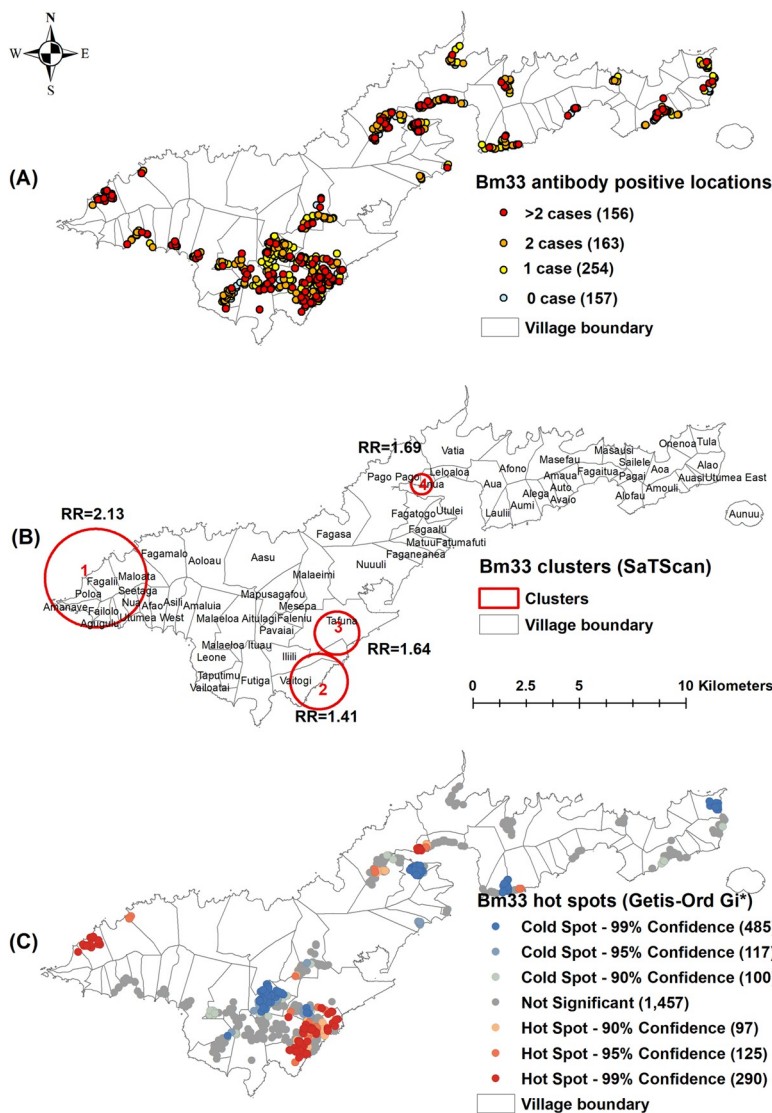

**Fig 6.** A) Spatial distribution of Bm33 antibody in survey locations, B) Spatial clusters of antibody positive survey locations (SaTScan), C) Hotspots of antibody positive survey locations (Getis-Ord Gi*), American Samoa 2016. Base layers from (https://www.diva-gis.org/Data).

**Table 1. Summary table of intra-cluster correlation (ICC) coefficients (adjusted for age and sex) for lymphatic filariasis infection markers by village and household levels, American Samoa 2016.**

| Tests | ICC coefficient (95% CI) | |
|---|---|---|
| | Household | Village |
| Antigen | 0.59 (0.45–0.71) | 0.17 (0.08–0.33) |
| Microfilaria | 0.69 (0.45–0.86) | 0.30 (0.10–0.61) |
| Wb123 Ab | 0.27 (0.20–0.36) | 0.11 (0.06–0.21) |
| Bm14 Ab | 0.33 (0.23–0.44) | 0.17 (0.09–0.29) |
| Bm33 Ab | 0.20 (0.14–0.28) | 0.10 (0.05–0.19) |

*ICC- intra-cluster correlation; **CI**- confidence interval

**Table 2. Parameters of significant spatial autocorrelation for lymphatic filariasis antigen, microfilaria and antibodies (Bm14, Bm33 and Wb123), American Samoa 2016.**

| Spatial parameters | Antigen | Microfilaria | Wb123 Ab | Bm14 Ab | Bm33 Ab |
|---|---|---|---|---|---|
| Partial sill | 0.03 | 0.01 | 0.45 | 0.12 | 0.93 |
| Range (degrees)* | 0.0051 | 0.0019 | 0.0036 | 0.0049 | 0.0020 |
| Range (meters) | 562 | 207 | 397 | 548 | 220 |
| Nugget | 0.19 | 0.03 | 0.76 | 0.44 | 1.39 |
| Percentage of variance due to spatial dependence (%) | 14 | 26 | 37 | 21 | 40 |
| Nugget/sill (%) | 86 | 75 | 63 | 79 | 60 |

*One decimal degree at the equator is approximately 111Km

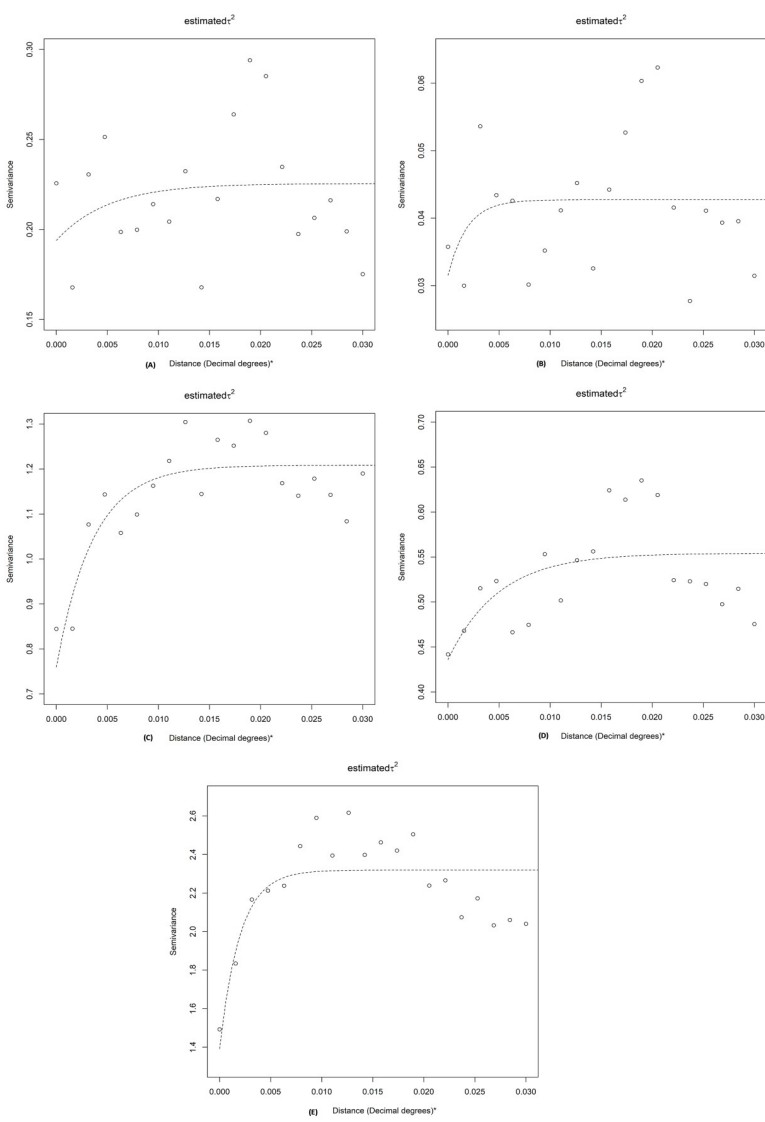

**Fig 7.** Semivariograms of spatial autocorrelation of A) lymphatic filariasis Ag, B) Mf, C) Wb123 Ab, D) Bm14 Ab and E) Bm33 Ab, American Samoa 2016. (One decimal degree at the equator is approximately 111Km).

**Table 3. Summary statistics from SaTScan using Bernoulli model for identifying significant clusters of lymphatic filariasis infection markers (microfilaria, antigen and antibodies), American Samoa 2016.**

| Parameters | Antigen | Microfilaria | Anti-filarial antibodies | | | | | | | |
|---|---|---|---|---|---|---|---|---|---|---|
| | | | Wb123 | | Bm14 | | Bm33 | | | |
| | Cluster 1 Fagali'i | Cluster 1 Fagali'i | Cluster 1 Fagali'i | Cluster 2 Vaitogi | Cluster 1 Fagali'i | Cluster 2 Vaitogi | Cluster 1 Fagali'i | Cluster 2 Vaitogi | Cluster 3 Ili'ili-Tafuna | Cluster 4 Pago Pago-Anua |
| Radius of scanning window (km) | 0.44 | 1.85 | 2.31 | 0.27 | 2.31 | 0.33 | 2.31 | 1.30 | 1.01 | 0.47 |
| Population inside window | 39 | 44 | 98 | 48 | 94 | 53 | 94 | 346 | 61 | 46 |
| Number of positive cases inside window (% positive) | 26 (66.7%) | 14 (31.8%) | 79 (80.6%) | 32 (66.7%) | 64 (68.1%) | 21 (39.6%) | 88 (93.6%) | 212 (61.3%) | 45 (73.8%) | 35 (76.1%) |
| Expected cases inside window | 1.97 | 0.56 | 25.10 | 12.29 | 12.32 | 6.94 | 42.90 | 157.9 | 27.84 | 20.99 |
| RR inside window for positive case | 16.10 | 41.46 | 3.43 | 2.68 | 6.13 | 3.15 | 2.13 | 1.41 | 1.64 | 1.69 |
| Log likelihood ratio | 55.89 | 37.04 | 68.06 | 18.16 | 80.18 | 11.92 | 51.97 | 19.56 | 10.18 | 9.0 |
| P value | <0.0001 | <0.0001 | <0.0001 | <0.0001 | <0.0001 | 0.0025 | <0.0001 | <0.0001 | 0.015 | 0.044 |

**RR**- relative risk

Yellow–Cluster 1, Blue–Cluster 2, Orange–Cluster 3, Green–Cluster 4

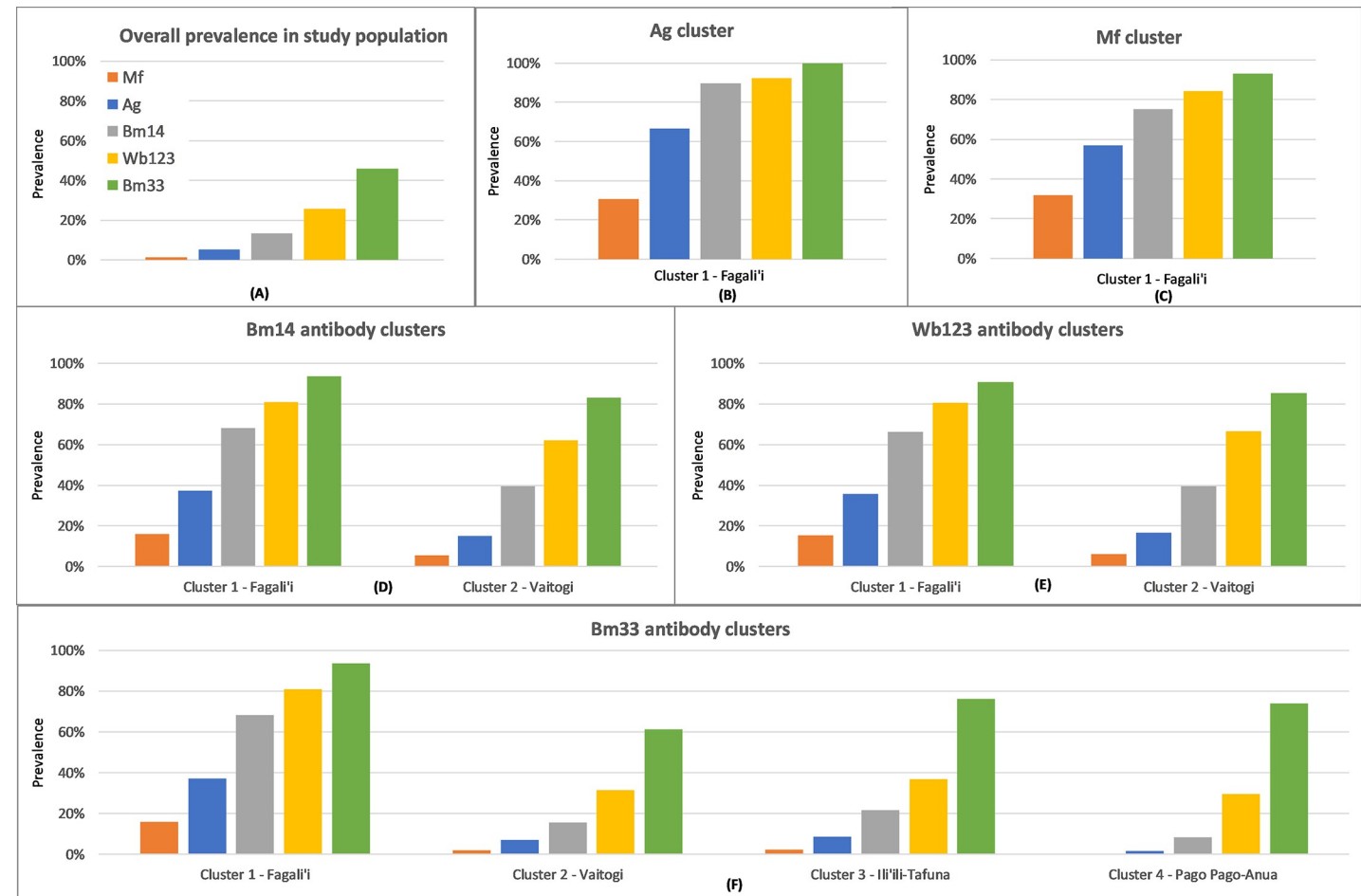

**Fig 8.** Prevalence of lymphatic filariasis Ag, Mf, and antibodies in A) the overall study population, and within SaTScan clusters of B) Ag, C) Mf, D) Wb123 Ab, E) Bm14 Ab, and F) Bm33 Ab, American Samoa 2016.

explored in this study, yielding the most detailed output in terms of spatial resolution and risk stratification.

The finding of higher ICCs at household than village level suggests that transmission was more likely to occur between household members than other village inhabitants. This is plausible because *Aedes polynesiensis*, the main vector for LF in America Samoa, has a short flight range of ~100 meters [44]. These results corroborate our findings in neighbouring Samoa, where ICC for Ag and Mf were higher for households (0.46 and 0.63) compared to PSUs (0.18 and 0.12) [45]. Strong clustering of infected persons within households suggests that household members of Ag-positive and Mf-positive persons should be offered testing and/or treatment as part of surveillance activities [46,47]. There is currently insufficient information about what the presence of each Ab means in terms of stage of infection or infectivity, and there are no recommendations to provide treatment based on Ab status alone. Further studies are required before recommending testing and/or treatment of household members of Ab-positive persons.

Semivariograms identified significant spatial dependency for all infection markers. This result differs from previous spatial analyses using samples collected in 2010, which identified spatial dependency for Ag (measured by Og4C3 ELISA) and Wb123 Ab, but not Bm14 Ab [19]. Mf slides were not assessed in the 2010 study and Bm33 Ab was not measured. In 2010, the estimated cluster sizes were larger for Ag (1.2 to 1.5 km) and smaller for Wb123 Ab (60 m). There are a number of potential explanations for the differences in results between the two time points. Semivariograms provide a global measure of spatial dependency (without location), so differences in cluster size may therefore not be comparable. Different sampling methods were used in 2010 (spatial sampling strategy that included all villages) and 2016 (a village cluster survey of selected villages). Another possible explanation is changes in the spatial distribution from a time of relatively low prevalence in 2010 (0.8% for Ag, 8.1% for Wb123 Ab, and 17.9% for Bm14 Ab) to a time of resurgence in 2016 (5.1% for Ag, 25.6% for Wb123 Ab, 13.1% for Bm14 Ab). This dramatic increase in prevalence could have affected spatial patterns and the size of clusters. Spatial distribution may continue to change as prevalence changes, e.g. reduction in prevalence after MDA, or increasing prevalence of persistent resurgence.

Both local spatial methods (SaTScan and Getis-Ord Gi*) confirmed clustering and hotspots of all infection markers in a previously identified hotspot around Fagali'i, in the far west of Tutuila (cluster 1) [18,19]. Getis-Ord Gi* classified multiple locations in Fagali'i as hotspots with 99% confidence for all markers. High prevalence of Mf in Fagali'i (31.8%, 14 Mf-positives out of 44 within the SaTScan window) strongly suggests active transmission. Fagali'i was first identified as a potential hotspot through a serological study of samples collected in 2010 [19] for a population proportionate survey of American Samoa. Repeat surveys in 2014 and 2016 confirmed very high Ag prevalence [18,21]. MDA was conducted in American Samoa from 2000 to 2006. Based on the results of these cross-sectional studies (2010, 2014, 2016), it was not possible to determine if transmission in Fagali'i had been interrupted by the rounds of MDA in the early-mid 2000s but restarted later due to reintroduction of parasites to the village, or whether transmission was never interrupted by MDA.

Both local spatial methods also identified clusters or hotspots in a previously known hotspot around Ili'ili-Vaitogi-Futiga (cluster 2). Getis-Ord Gi* identified hotspots of all infection markers in this area, while SaTScan identified significant clusters of the three Abs. Although SaTScan did not identify any clusters of Ag and Mf in the area, Fig 8 shows that Ag prevalence was high within all Ab clusters. Furthermore, Mf-positive persons were present in all Ab clusters, confirming active transmission. SaTScan's failure to identify an Ag cluster in this area reflects SaTScan's use of RR for determining clustering, i.e. when overall prevalence in the study area is high (5.1% for Ag), prevalence inside a scanning window may have to be

extremely high for the difference to be significant. It is therefore important to note that absence of clustering on SaTScan does not mean absence of a problem. Also, SaTScan analyses are two dimensional (based on xy coordinates), and only consider Euclidean distance, e.g. it would not take into account large valleys between adjacent villages, or long road distances between seemingly nearby locations on a map. These examples shows that from an epidemiological perspective, clustering of Abs could be useful for identifying Ag-positive and Mf-positive persons.

SaTScan and Getis-Ord Gi* identified two other clusters of Bm33 Ab (Figs 2C, 6B and 6C). One is in Ili'ili-Tafuna (cluster 3), east of our previously identified hotspot in Ili'Ili/Vaitogi/ Futiga. The other is a newly identified cluster in the Pago Pago-Anua area (cluster 4). Near the Bm33 Ab cluster in the Pago Pago-Anua area identified by SaTScan (cluster 4) (Fig 6B), Getis-Ord Gi* identified hotspots of all infection markers (Figs 2C, 3C, 4C, 5C, and 6C). The presence of multiple Mf-positive persons in these areas (Figs 2A and 3A) strongly suggests ongoing transmission. Ili'ili and Pago Pago were locations of schools where Ag-positive children were identified in TAS [17,21].

Getis-Ord Gi* identified hotspots of Wb123 Ab in the small villages of Nua-Seetaga-Asili (Fig 4C). Smaller Ag hotspots were also identified in this area (Fig 2C). TAS-3 identified two Ag-positive children in the elementary school in Nua, the closest school for children living in Fagali'i [21]. The absence of Bm14 and Bm33 Ab hotspots in this area may indicate that transmission was more recent than those in clusters 2 and 3 because these antibodies persist for years.

SaTScan and Getis-Ord Gi* further confirmed Fagali'i and Ili'ili-Vaitogi as likely hotspots. All four SaTScan clusters found in this study were either known hotspots from previous studies [18,48], and/or locations where Ag-positive children were identified in TAS-3 in 2016. The Wb123 Ab hotspot identified by Getis-Ord Gi* around Nua-Seetaga-Asili is also a known area of concern, where two Ag-positive children were identified in TAS-3. For the SaTScan results, high prevalence of all infection markers (including Ag and Mf) within most clusters (Fig 8) suggest that they represent areas of ongoing transmission. The significance of Getis-Ord Gi* hotspots around the Malaeimi is unclear, as we do not have any prior signals from this area.

A strength of this study is the availability of data for five infection makers, representing different probabilities and stages of infection. Different patterns were seen with each infection marker which may represent clusters and hotspots that are emerging, active or formerly active. Further work and longitudinal follow up of individuals are needed to fully appreciate and interpret the results provided by each infection marker.

This study has limitations that should be noted. Firstly, the 2016 survey did not include all villages so some hotspots may have been missed. Secondly, we used household locations of individuals for all analyses, and did not account for work or school locations [49] even though American Samoa has day-biting mosquito vectors, and those working outdoors are more likely to be infected with LF [20,50]. Despite these limitations, our study was able to identify and verify areas of high transmission.

This study has contributed to the knowledge of spatial heterogeneity in LF transmission, and shown that spatial analysis can be used to identify clusters and hotspots. Our findings highlight the importance of understanding the differences between spatial analytical methods. The choice of methods will depend on the purpose of the analysis, and using a combination of methods (as we have done in this study) should also be considered. The programmatic value of the results will vary depending on the stage of elimination and also differ between countries depending on programme and history of endemicity. It is likely that spatial clustering will become more intense as infection reaches very low levels, so cluster analysis may help to delineate and prioritise areas of ongoing transmission before infection can spread and lead to more

widespread resurgence. Given the greater sensitivity of Abs for cluster identification compared to Ag and Mf, Ab markers could be an additional tool for identifying areas for test and treat programmes. However, further studies of the relationships between Abs, Ag and Mf over space and time are needed before recommendations about more sensitive and specific diagnostic strategies or markers can be made.

The spatial heterogeneity of LF, even on a very small island in American Samoa, suggest that infection risk is likely to be driven by highly localised and spatially-explicit factors, e.g. human behaviour, mosquito distribution and density, previous MDA coverage, or a combination of these factors. Further spatial analyses, e.g. Bayesian geostatistics, may be able to identify drivers of transmission that vary over space, and use this information to produce predictive risk maps that include outputs for unsampled locations [51]. Risk maps could be used by LF programmes for prioritising or intensifying LF elimination efforts in high-risk areas, e.g. health promotion to maximise MDA coverage, vector control, targeted testing and/or treating in communities and schools, and more intensive surveillance.

## Conclusion

Our study further confirmed previously known hotspots of LF transmission in American Samoa and identified other potential hotspots that warrant further investigation. We demonstrated the utility of non-spatial and spatial methods of investigating spatial clustering and hotspots, the differences in information provided by each method, and the value of triangulating results from multiple methods. We also showed the benefit of using multiple infection markers for cluster and hotspot analyses.

## Supporting information

**S1 Table. Number and percentage of households (n = 750) with participants with positive infection markers for lymphatic filariasis, American Samoa 2016.**
(DOCX)

**S2 Table. Number and percentage of unique survey locations (n = 730) with participants with positive infection markers for lymphatic filariasis, American Samoa 2016.**
(DOCX)

**S3 Table. Moran's I statistics (measure of autocorrelation) for five infection markers of lymphatic filariasis, American Samoa 2016.**
(DOCX)

## Acknowledgments

We would like to acknowledge the hard work of all our field team members, particularly Ms Paeae Tufono. We would also like to thank Ms Mary Matai'a of the American Samoa Department of Health for her assistance with logistics and laboratory testing of specimens. We also thank Dr Mark Schmaedick at the American Samoa Community College for generously allowing us to use his laboratories. We are also grateful to all the school principals and teachers, and village mayors and chiefs for their assistance in conducting the fieldwork. We thank Dr Kei Owada and A/Prof Ricardo Soares Magalhaes for providing technical advice on semivariograms. We also thank Dr Kimberly Won and Keri Robinson (Division of Parasitic Diseases and Malaria, US Centers for Disease Control and Prevention) for conducting the laboratory testing for antibodies.

## Author Contributions

**Conceptualization:** Kinley Wangdi, Colleen L. Lau.

**Data curation:** Kinley Wangdi, Meru Sheel, Patricia M. Graves, Colleen L. Lau.

**Formal analysis:** Kinley Wangdi, Patricia M. Graves.

**Funding acquisition:** Colleen L. Lau.

**Investigation:** Colleen L. Lau.

**Methodology:** Kinley Wangdi, Meru Sheel, Patricia M. Graves, Colleen L. Lau.

**Project administration:** Meru Sheel, Saipale Fuimaono, Colleen L. Lau.

**Resources:** Saipale Fuimaono.

**Supervision:** Meru Sheel, Saipale Fuimaono, Patricia M. Graves, Colleen L. Lau.

**Validation:** Kinley Wangdi, Meru Sheel, Patricia M. Graves, Colleen L. Lau.

**Visualization:** Kinley Wangdi, Colleen L. Lau.

**Writing – original draft:** Kinley Wangdi, Colleen L. Lau.

**Writing – review & editing:** Kinley Wangdi, Meru Sheel, Saipale Fuimaono, Patricia M. Graves, Colleen L. Lau.

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
