## [Decision Letter · Decision Letter 0]

7 Sep 2021

Dear Dr. Wangdi,

Thank you very much for submitting your manuscript "Lymphatic filariasis in 2016 in American Samoa: Identifying clustering and hotspots using non-spatial and three spatial analytical methods" for consideration at PLOS Neglected Tropical Diseases. As with all papers reviewed by the journal, your manuscript was reviewed by members of the editorial board and by several independent reviewers. In light of the reviews (below this email), we would like to invite the resubmission of a significantly-revised version that takes into account the reviewers' comments. 

We cannot make any decision about publication until we have seen the revised manuscript and your response to the reviewers' comments. Your revised manuscript is also likely to be sent to reviewers for further evaluation.

Sincerely,

Subash Babu

Associate Editor

Robert Reiner

Deputy Editor

Reviewer's Responses to Questions

**Key Review Criteria Required for Acceptance?**

**Methods**

-Are the objectives of the study clearly articulated with a clear testable hypothesis stated?

-Is the study design appropriate to address the stated objectives?

-Is the population clearly described and appropriate for the hypothesis being tested?

-Is the sample size sufficient to ensure adequate power to address the hypothesis being tested?

-Were correct statistical analysis used to support conclusions?

-Are there concerns about ethical or regulatory requirements being met?

Reviewer #1: The authors present the spatial and non-spatial methods in a clear manner for readers who may not be familiar with such analyses. The dataset is rich with household GPS coordinates, and antigen, microfilariae, and antibody data for ~2500 participants from 30 villages in American Samoa.

Reviewer #2: Some additional analysis is required to add value to the manuscript

Reviewer #3: Yes to all

**Results**

-Does the analysis presented match the analysis plan?

-Are the results clearly and completely presented?

-Are the figures (Tables, Images) of sufficient quality for clarity?

Reviewer #1: The results presented in this manuscript are clear and follow the analysis plan described in the methods. The figures and tables match the data presented in the results section of the paper and are presented quite nicely.

-There is one figure, Figure 6, which is out of order at the end of the manuscript where all figures are presented.

Reviewer #2: Additional tables requested

Reviewer #3: Yes study results match with the statistical analysis plan. Specific comments are given in the attached Reviewer's comments.

**Conclusions**

-Are the conclusions supported by the data presented?

-Are the limitations of analysis clearly described?

-Do the authors discuss how these data can be helpful to advance our understanding of the topic under study?

-Is public health relevance addressed?

Reviewer #1: The authors' interpretations of the results are thorough and the conclusions are supported by the data presented. An added strength is the comparison to the known hotspots from the previous studies. The authors clearly present the limitations of the study and how the analytical methods can be used.

In the authors summary, the authors note that 'embedding the tools used in this study into operational planning for LF surveillance may identify ongoing transmission not identified through routine surveys'. I think it would be helpful to describe a bit about the feasibility of this (i.e. do country programs generally have the capacity to do such analyses). Also, I would be curious to know how the American Samoa LF elimination program plans to use these data if there is a plan to do so.

Reviewer #2: Needs to be strengthened

Reviewer #3: Yes to all. The conclusions are supported with data. Study limitations are clearly given and discussed how further methodological improvments would be hhelpful to have a better understanding the topic of the sudy.

**Editorial and Data Presentation Modifications?**

Reviewer #1: • Line 72 states 120mil people are infected with LF, but a recent WER updated this number to 51.4 million as of 2018. LF- WER9543 Global programme to eliminate lymphatic filariasis: progress report, 2019

Reviewer #2: (No Response)

Reviewer #3: Minor comments

1) Line 39: insert “spatial” before “clustering”

2) Line 98: please mention the drug regimen used in the MDA rounds.

3) L137-139: The description given in this para gives the impression that the data used in the present study were collected based on both the WHO recommended TAS and multistage cluster survey as described by Sheel et al. Please make it clear that the study by Sheel et al (2018) collected data in 2016 via (i) WHO recommended systematic school-based TAS and (ii) two-stage (rather than multi-stage, can be specific to say as two-stage: stage 1: cluster (village) and stage 2: household) community-based household survey. The authors have to clearly state that the present study used the data from community-based household survey. 

4) Lines 176-182: The para may be shortened by explaining the method and its usefulness for spatial dependency. The sentences between lines 176-182 may be deleted as they give more of description about semivariogram. The sentence starting from “Outputs from …..location of hotspots” may be moved to the end of next para after line 199.

5) L184-185: The authors can also calculate the Local Moran’s I, a local indicator of spatial association (LISA), to identify the local spatial clusters. It measures the degree of spatial autocorrelation at each specific location (Anselin, 1995: Local indicators of spatial association – LISA. Geogr. Anal., 27, 93–115) as it compares only the neighboring value rather than the overall mean. A high positive local Moran’s I value implies the target value is similar to its neighborhood, and then the locations are spatial clusters, which include high–high clusters (high values in a high value neighborhood) and low–low clusters (low values in a low value neighborhood). Meanwhile, a high negative local Moran’s I value implies a potential spatial outlier, which is obviously different from the values of its surrounding locations.

6) Line 202: “locations of significant clusters”. Whether the data for this analysis is positivity at household level? What does locations refer to? Is it location of households? Please clarify.

7) Lines 249 & 261 -271: The 2016 community survey include only 2507 persons (including 11 invalid FTS tests) from 711 households (hh) in 30 PSUs (please see Shheel et al.2018). Whereas in the present study the corresponding values were given as 2671 participants from 750 hhs (730 unique locations) in 32 PSUs in 30 villages. The values differ from the original study results. Similarly, the infection results presented in Lines 261-271 are different from that presented in the original study. Please clarify.

- Please cite PLOS NTD 2020, instead of ‘medRxIV’ for ref. 20. 

- Delete the repeat citation ref. 34, which is same as 20.

- Please check the citation 34: should it be 19?

8) Lines 290-291: How was the statistical significance of spatial-dependency determined? Please mention the statistical test used under Methods and its results in the Table or text. 

9) Table 2 may be deleted, as the semivariogram (Fig.7) describes the spatial dependency. All relevant data are already given in the text.

10) Table 3: Title is not clear and should also include “Mf”. Retitle it as “Summary statistics from SaTScanTM using Bernoulli model for identifying significant clusters of microfilaria, antigen and antibody positives”.

11) Please explain how the RRs inside thw window were calculated for each significant cluster. More specifically, what is the reference group for calculating the RR.

12) Fig 8. The prevalence of Ag and Mf within all significant clusters are higher than that of antibody clusters. This is somewhat unexpected. Please explain.

13) Lines 366-381: The semi-variogram analyses measures spatial auto-correlation of the outcome interest for all pairs of observations within each lag. The estimated cluster sizes are expected to be independent of sampling methods, provided the samples have spatial representation in the study area. Therefore, the larger cluster sizes for Ag and smaller for Wb123 estimated in 2010 study compared to the present study could not be attributed to difference in sampling methods. But the temporal changes in Ag-prevalence could have impacted the cluster sizes. Another explanation could be that the tests used for detecting antigen were different (og4C3 vs FTS): FTS is more sensitive than Og4C3. Therefore, the above explanation may be applicable to Ag but not for Wb123 Ab as it is expected to be for a longer period even in the absence of resurgence and hence though an increase in Ab prevalence is expected if there is any resurgence but not a reduction. The reason for smaller cluster size for Wb123 in the 2010 survey compared to that in 2016 is difficult to explain. I would suggest the authors to revise the text in light of the above, if agreeable or else need clarification.

14) Line 470: references to earlier related work on spatial risk map may be added (e.g. Sabesan et al.(2013). Vector borne and zoonotic diseases 13(9). DOI: 10.1089/vbz.2012.1238).

15) Discussion is too lengthy to read, the authors may consider shortening the discussion.

**Summary and General Comments**

Reviewer #1: This is a well written manuscript that illustrates how spatial and non-spatial analysis can be used to assess clustering of biomarkers, identify hotspots, and compares the various methods. I found the authors' step by step description of the different analyses in the methods section very helpful and appreciated the level of detail included.

Reviewer #2: Need improvement. Write up is a bit confusing at certain places.

Reviewer #3: General remarks:

This ms aims to compare the usefulness of different spatial (Kuldorf’ scan statistic, Getis-Ord G* & variogram) and non-spatial (ICC) methods to characterize the clustering and hotspots of LF in American Samoa, which passed school-based TAS 1 and 2 after 7 rounds of MDA. Subsequently in 2016, a separate operational research study which compared the results of school-based TAS-3 and a community-based survey indicated resurgence of LF after passing TAS 2. In the present study, the authors using the data from 2016 community-based survey, demonstrated the usefulness of the spatial and non-spatial analytical methods for identifying LF clusters and hotspots in terms of Ag, and three antibody markers (Wb123, Bm14 &Bm33) in American Samoa. The ms is well written, with clear objectives, applied appropriate statistical methods and the results are presented with appropriate tables and figures. The Discussion is too lengthy to read, it is desirable to shorten it. Altogether, I would recommend the ms for publication after a revision based on the following minor comments.

PLOS authors have the option to publish the peer review history of their article (what does this mean?). If published, this will include your full peer review and any attached files.

Reviewer #1: No

Reviewer #2: No

Reviewer #3: No
---

## [Decision Letter · Decision Letter 1]

15 Dec 2021

Dear Dr. Wangdi,

Thank you very much for submitting your manuscript "Lymphatic filariasis in 2016 in American Samoa: Identifying clustering and hotspots using non-spatial and three spatial analytical methods" for consideration at PLOS Neglected Tropical Diseases. As with all papers reviewed by the journal, your manuscript was reviewed by members of the editorial board and by several independent reviewers. The reviewers appreciated the attention to an important topic. Based on the reviews, we are likely to accept this manuscript for publication, providing that you modify the manuscript according to the review recommendations. 

Sincerely,

Subash Babu

Associate Editor

Robert Reiner

Deputy Editor

Reviewer's Responses to Questions

**Key Review Criteria Required for Acceptance?**

**Methods**

-Are the objectives of the study clearly articulated with a clear testable hypothesis stated?

-Is the study design appropriate to address the stated objectives?

-Is the population clearly described and appropriate for the hypothesis being tested?

-Is the sample size sufficient to ensure adequate power to address the hypothesis being tested?

-Were correct statistical analysis used to support conclusions?

-Are there concerns about ethical or regulatory requirements being met?

Reviewer #1: I think the addition of the 'infection marker' section is a very helpful addition for readers.

Reviewer #2: Yes

Reviewer #3: Yes to all, except a methodological suggestion to test the significance of "spatial dependency" Please see attachment for details

**Results**

-Does the analysis presented match the analysis plan?

-Are the results clearly and completely presented?

-Are the figures (Tables, Images) of sufficient quality for clarity?

Reviewer #1: Happy to see the addition of p-values in the results.

Reviewer #2: Yes, adding one more table could be useful.

As suggested in the earlier comments, fitting distributions (Poisson/Negative Binomial) to the no. of positives at the household level for different indicators also is a non-spatial analysis to indicate if the distribution of cases are random or clustered.

Reviewer #3: Yes to all with minor revisions in the Results based on the suggested method (indicated above) in Table 2 and Discussion.

**Conclusions**

-Are the conclusions supported by the data presented?

-Are the limitations of analysis clearly described?

-Do the authors discuss how these data can be helpful to advance our understanding of the topic under study?

-Is public health relevance addressed?

Reviewer #1: none

Reviewer #2: The conclusion may be strengthened in terms of how these analytical methods would be helpful in post elimination phase where the cases would have drastically come down and may not exhibit any clusters..Also on the indicator that could be used not to miss out any cluster during the post validation phase.

Reviewer #3: Yes to all, but depending on the revisions based on the spatial dependency analysis suggested above

**Editorial and Data Presentation Modifications?**

Reviewer #1: none

Reviewer #2: None

Reviewer #3: Please see attachment for minor corrections

**Summary and General Comments**

Reviewer #1: The authors addressed my initial minor feedback from the first review. I think the manuscript is strong and adds to ongoing work on geospatial analysis and clustering of infection, both of which are important as LF elimination programs are reaching the last mile.

Reviewer #2: This manuscript is important particularly as it compares different spatial clustering techniques.

Reviewer #3: The authors have revised the manuscript addressing most of my concerns satisfactorily. I congratulate the authors for demonstrating the potential usefulness of different non-spatial and spatial methods for characterizing LF clusters and hotspots based on different LF indicators. I have a few more questions which I believe is important before the ms is accepted for publication.

PLOS authors have the option to publish the peer review history of their article (what does this mean?). If published, this will include your full peer review and any attached files.

Reviewer #1: No

Reviewer #2: No

Reviewer #3: Yes: Dr. Swaminathan Subramanian

Figure Files:

Data Requirements:

Reproducibility:

References

---

## [Decision Letter · Decision Letter 2]

15 Feb 2022

Dear Dr. Wangdi,

We are pleased to inform you that your manuscript 'Lymphatic filariasis in 2016 in American Samoa: Identifying clustering and hotspots using non-spatial and three spatial analytical methods' has been provisionally accepted for publication in PLOS Neglected Tropical Diseases.

Best regards,

Subash Babu

Associate Editor

Robert Reiner

Deputy Editor

Reviewer's Responses to Questions

**Key Review Criteria Required for Acceptance?**

**Methods**

-Are the objectives of the study clearly articulated with a clear testable hypothesis stated?

-Is the study design appropriate to address the stated objectives?

-Is the population clearly described and appropriate for the hypothesis being tested?

-Is the sample size sufficient to ensure adequate power to address the hypothesis being tested?

-Were correct statistical analysis used to support conclusions?

-Are there concerns about ethical or regulatory requirements being met?

Reviewer #3: The authors have addressed all the methodological issues raised on the previous version, particularly the methodology related to 'spatial dependency'.

**Results**

-Does the analysis presented match the analysis plan?

-Are the results clearly and completely presented?

-Are the figures (Tables, Images) of sufficient quality for clarity?

Reviewer #3: The authors have revised the results on spatial dependency as per the comments given in the previous version.

**Conclusions**

-Are the conclusions supported by the data presented?

-Are the limitations of analysis clearly described?

-Do the authors discuss how these data can be helpful to advance our understanding of the topic under study?

-Is public health relevance addressed?

Reviewer #3: Yes

**Editorial and Data Presentation Modifications?**

Reviewer #3: Very minor:

LL 107-109: Insert citation to the statement “The term “resurgence” was used to indicate significant increase in infection prevalence to levels above target thresholds”.

Table 1: "........infection markers village and household levels," INSERT "by" before village

**Summary and General Comments**

Reviewer #3: The authors addressed all my comments from the first review. I think the manuscript has improved a lot and adds value to the work on geospatial analysis and clustering of infection.

PLOS authors have the option to publish the peer review history of their article (what does this mean?). If published, this will include your full peer review and any attached files.

Reviewer #3: No

---

## [Editor Report · Acceptance letter]

22 Mar 2022

Dear Dr. Wangdi,

We are delighted to inform you that your manuscript, "Lymphatic filariasis in 2016 in American Samoa: Identifying clustering and hotspots using non-spatial and three spatial analytical methods," has been formally accepted for publication in PLOS Neglected Tropical Diseases.

Best regards,

Shaden Kamhawi

co-Editor-in-Chief

Paul Brindley

co-Editor-in-Chief
